complexity

football, network, perdition, centrality

**Author for correspondence:**
Taha Yasseri
e-mail: taha.yasseri@ucd.ie

# Football is becoming more predictable; network analysis of 88 thousand matches in 11 major leagues

Victor Martins Maimone[1] and Taha Yasseri[1,2,3,4]

[1]Oxford Internet Institute, University of Oxford, Oxford OX1 3JS, UK
[2]Alan Turing Institute, London NW1 2DB, UK
[3]School of Sociology, and [4]Geary Institute for Public Policy, University College Dublin, Dublin 4, Ireland

TY, 0000-0002-1800-6094

In recent years, excessive monetization of football and professionalism among the players have been argued to have affected the quality of the match in different ways. On the one hand, playing football has become a high-income profession and the players are highly motivated; on the other hand, stronger teams have higher incomes and therefore afford better players leading to an even stronger appearance in tournaments that can make the game more imbalanced and hence predictable. To quantify and document this observation, in this work, we take a minimalist network science approach to measure the predictability of football over 26 years in major European leagues. We show that over time, the games in major leagues have indeed become more predictable. We provide further support for this observation by showing that inequality between teams has increased and the home-field advantage has been vanishing ubiquitously. We do not include any direct analysis on the effects of monetization on football's predictability or therefore, lack of excitement; however, we propose several hypotheses which could be tested in future analyses.

## 1. Introduction

Playing football is, arguably, fun. So is watching it in stadiums or via public media, and to follow the news and events around it. Football is worthy of extensive studies, as it is played by roughly 250 million players in over 200 countries and dependencies, making it the world's most popular sport [1]. European leagues alone are estimated to be worth more than £20 billion [2]. The sport itself is estimated to be worth almost 30 times as much [3]. In electronic supplementary material, figure S1, the combined revenue for the five major European

leagues over time are shown. The overall revenues have been steadily increasing over the last two decades [4]. It has been argued that the surprise element and unpredictability of football is the key to its popularity [5]. A major question in relation to such a massive entertainment enterprise is if it can retain its attractiveness through surprise element or, due to significant recent monetizations, it is becoming more predictable and hence at the risk of losing popularity? To address this question, a first step is to establish if the predictability of football has indeed been increasing over time. This is our main aim here; to quantify predictability, we devise a minimalist prediction model and use its performance as a proxy measure for predictability.

There has previously been a fair amount of research in statistical modelling and forecasting in relation to football. The prediction models are generally either based on detailed statistics of actions on the pitch [6–9] or on a prior ranking system which estimates the relative strengths of the teams [10–12]. Some models have considered team pairs attributes such as the geographical locations [7,13,14], and some models have mixed these approaches [15,16]. A rather new approach in predicting performance is based on machine learning and network science [17,18]. Such methods have been used in relation to sport [19–21] and particularly football [22,23]. Most of the past research in this area, however, either focuses on inter-team interactions and modelling player behaviour rather than league tournament's results prediction, or are limited in scope—particularly, they rarely take an historical approach in order to study the game as an evolving phenomenon [24–26]. This is understandable in light of the fact that most of these methods are data-thirsty and therefore not easy to use in an historical context where extensive datasets are unavailable for games played in the past.

In the present work, we use a network science approach to quantify the predictability of football in a simple and robust way without the need for an extensive dataset and by calculating the measures in 26 years of 11 major European leagues we examine if predictability of football has changed over time.

# 2. Results and discussion

The predictive model and the method of quantifying its performance are presented in the Data and methods section. After having assessed the models' validity and robustness, here we address if the predictability of football has been changing over time.

## 2.1. Predictability over time

In figure 1, we see the area under the curve (AUC) scores (a measure of the performance of the model) and smoothed fits to them per league over time, for the last 26 years (and in electronic supplementary material, figure S13 you see similar trends measured by Brier score). A positive trend in predictability is observed in most of the cases (England, France, Germany, The Netherlands, Portugal, Scotland and Spain) and, for the cases of the top leagues (England, Germany, Portugal and Spain), the graphical intuition is corroborated by a comparison between the earlier and later parts of their respective samples using the Student $t$-test. We compared the first 10 years with the last 10 years and reported the $p$-value for the test (under the null hypothesis that the expected values are the same). We also compared the two distributions under the Kolmogorov–Smirnov (KS) test to check whether they were similar (under the null hypothesis that they were). Both sets of $p$-values are reported in table 1. The remaining leagues display somewhat stable predictability throughout time. All leagues, notwithstanding, tend to converge towards 0.75 AUC.

## 2.2. Increasing inequality

In analysing the predictability of different leagues, we observe that predictability has been increasing for the richer leagues in Europe, whereas the set for which the indicator is deteriorating is composed mainly of peripheral leagues.

It seems football as a sport is emulating society in its somewhat 'gentrification' process, i.e. the richer leagues are becoming more deterministic because better teams win more often; consequentially, becoming richer; allowing themselves to hire better players (from a talent pool that gets internationally broader each year); becoming even stronger; and, closing the feedback cycle, winning even more matches and tournaments; hence more predictability in more professional and expensive leagues.

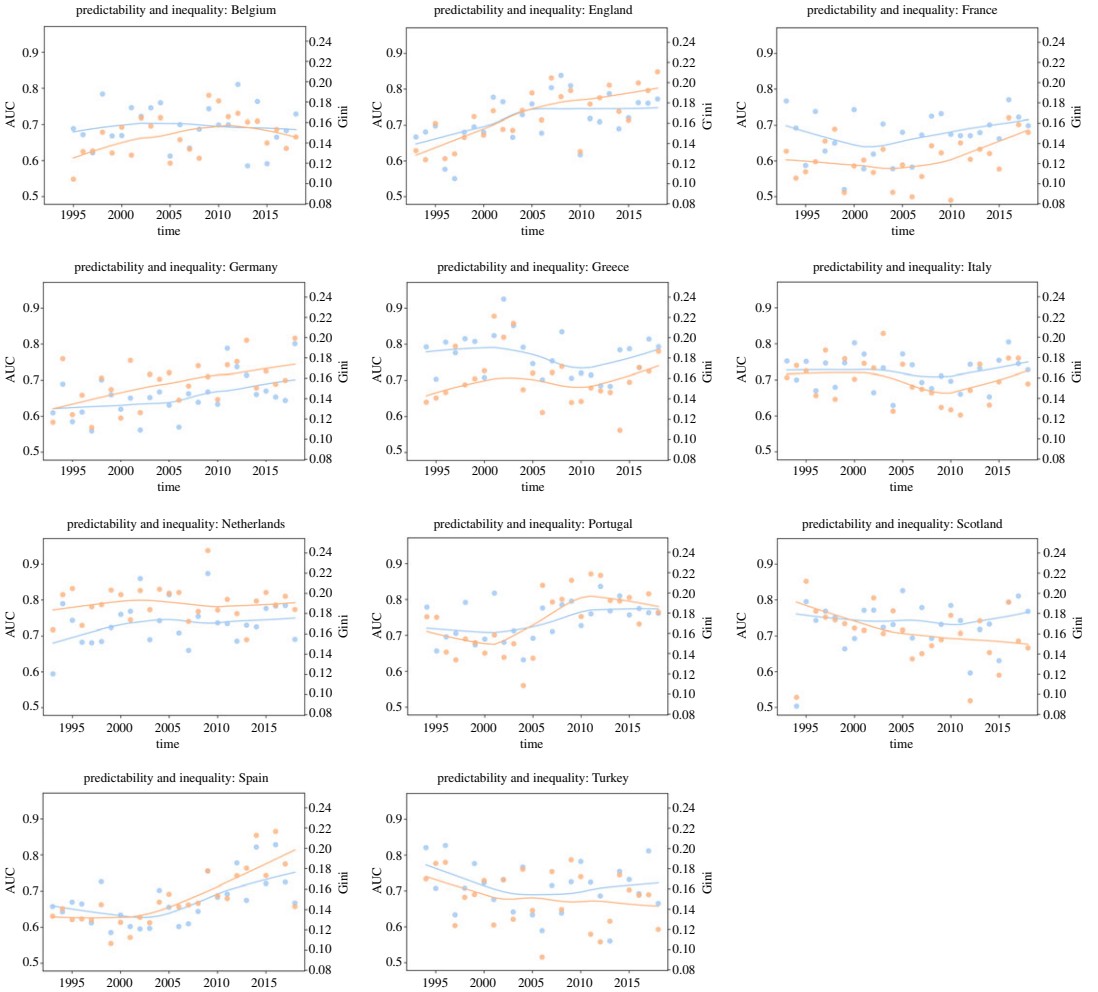

**Figure 1.** Time trends in AUC and Gini coefficient. Blue dots/lines depict AUC and are marked in the primary (left) *y*-axis; orange dots/lines depict the Gini coefficient and are marked in the secondary (right) *y*-axis; both lines are fitted through a locally weighted scatterplot smoothing (LOWESS) model.

**Table 1.** The *p*-values from comparing the average network model AUC, inequality coefficient (Gini) and the Elo-system AUC (ELO-AUC) for the two parts of each country's sample, under the null hypothesis that the expected values are the same using the Student *t*-test. We also present the *p*-values for the distributions under the KS test to check whether they were similar (under the null hypothesis that they were). *p*-values < 0.05 are in italics.

| country | AUC: KS | AUC: *t* | Gini: KS | Gini: *t* | ELO-AUC: KS | ELO-AUC: *t* |
|---|---|---|---|---|---|---|
| Belgium | 0.9985 | 0.7383 | 0.2558 | 0.0752 | 0.8690 | 0.4805 |
| England | 0.1265 | *0.0330* | *0.0126* | *0.0009* | *0.0443* | *0.0133* |
| France | 0.1265 | 0.1437 | 0.2999 | 0.3426 | 0.1265 | 0.0895 |
| Germany | 0.1265 | *0.0326* | 0.2999 | *0.0367* | 0.5882 | 0.0787 |
| Greece | 0.2558 | 0.0634 | 0.5361 | 0.1190 | *0.0079* | *0.0019* |
| Italy | 0.2999 | 0.7370 | *0.0443* | 0.0547 | 0.9992 | 0.8292 |
| Netherlands | 0.8978 | 0.7371 | 0.5882 | 0.9488 | 0.8978 | 0.8320 |
| Portugal | *0.0079* | *0.0044* | *0.0000* | *0.0000* | *0.0314* | *0.0071* |
| Scotland | 0.9985 | 0.9128 | *0.0314* | 0.0771 | 0.5361 | 0.2864 |
| Spain | *0.0443* | *0.0097* | *0.0005* | *0.0001* | *0.0029* | *0.0010* |
| Turkey | 0.9985 | 0.6429 | 0.8690 | 0.4610 | 0.2558 | 0.3108 |

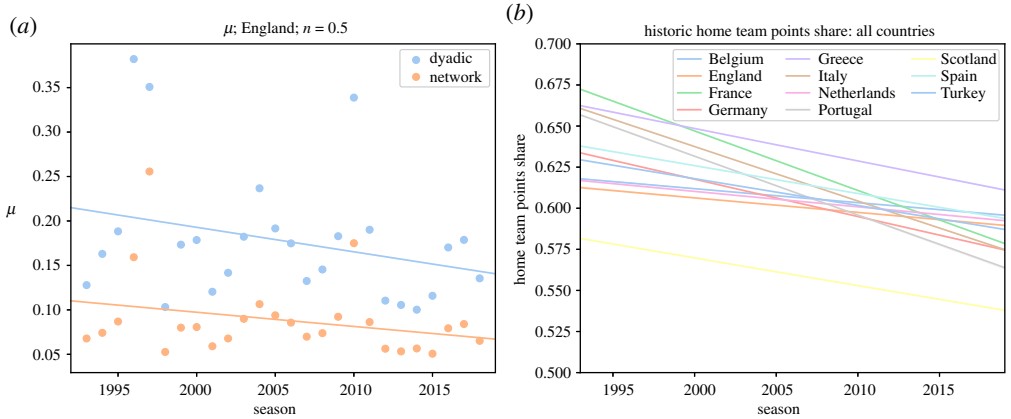

**Figure 2.** The decrease in home-field advantage. (*a*) Home-field advantage calculated by the two models. (*b*) The share of home points measured on historical data. The straight lines are linear fits. See the detailed graphs for each league in electronic supplementary material, figures S8–S11.

**Table 2.** AUC and Gini correlation by league.

| league | correlation |
|---|---|
| Spain | 0.874 |
| England | 0.823 |
| Germany | 0.805 |
| Scotland | 0.723 |
| Portugal | 0.694 |
| Turkey | 0.686 |
| Netherlands | 0.676 |
| Italy | 0.622 |
| France | 0.563 |
| Greece | 0.561 |
| Belgium | 0.413 |

To illustrate this trend, we use the Gini coefficient, proposed as a measure of inequality of income or wealth [27–29]. We calculate the Gini coefficient of a given league-season's distribution of points each team had at the end of the tournament. Figure 1 depicts the values for all the leagues in the database, comparing the evolution of predictability and the evolution of inequality between teams for each case. There is a high correlation between predictability in a football league and inequality among teams playing in that league, uncovering evidence in favour of the 'gentrification of football' argument. The correlation values between these two parameters are reported in table 2.

## 2.3. Home advantage and predictability

As described in the Data and methods section, our predictive model also quantifies the home-field advantage. We calculated the average amount of home-field advantage as measured in the prediction models ($\mu$ in equation (4.1)) for each season of the English Premier League and plotted it in figure 2. We see a decrease in home-field advantage over time (for the results for other leagues see electronic supplementary material, figures S2–S5.) However, one can calculate the home advantage directly from historical data by counting the total number of points that the home and away teams gained in each season. The trends in the share of home teams for different leagues are shown in figure 2 (for detailed diagrams of each league see electronic supplementary material, figures S2–S5).

It is clear that the home-field advantage is still present; however, it has been decreasing throughout time for all the leagues under study. In explaining this trend, we should consider factors involved in home-field advantage. Pollard counts the following as driving factors for the existence of home-field advantage [30]: crowd effects; travel effects; familiarity with the pitch; referee bias; territoriality; special tactics; rule factors; psychological factors; and the interaction between two or more of such factors. While some of these factors are not changing over time, increase in the number of foreign players, diminishing the effects of territoriality and its psychological factors, as well as observing that fewer people are going to stadiums, travelling is becoming easier, teams are camping in different pitches and players are accruing more international experience, can explain the reported trend; stronger (richer) teams are much more likely to win, it matters less where they play.

# 3. Conclusion

Relying on large-scale historical records of 11 major football leagues, we have shown that, throughout time, football is dramatically changing; the sport is becoming more predictable; teams are becoming increasingly unequal; the home-field advantage is steadily and consistently decreasing for all the leagues in the sample.

The prediction model in this work is designed to be self-contained, i.e. to solely consider the past results as supporting data to understand future events. Despite the strength in terms of time consistency, it does not rely on the myriad of data that are available nowadays, which is being extensively used by cutting-edge data science projects [31] and, obviously, the betting houses. In this sense, this work is limited by the amount of data and the sophistication of the models it considers; however, this is by design and in order to focus on the nature of the system under study rather than the predictive models.

Furthermore, this work is also limited by the sample size; football has been played in over 200 countries, out of which only 5.5% were included in our sample. Although the main reasons for that are of practical matters, given the hardship in obtaining reliable and time-consistent data, the events become very sparse for most of the leagues going back in time.

Future work should, as speculated in this work, try and assess: the role money is playing in removing the surprise element of the sport; expanding the sample barriers to include continent-level tournaments (such as the UEFA Champions League) and to look beyond the European continent; and ultimately, but not exhaustively, should test the money impact over predictability on different sports and leagues that—theoretically—should not be affected by it, namely leagues and sports that impose salary caps over their teams, such as the USA's professional basketball league (the NBA).

# 4. Data and methods

## 4.1. Data

In this study, each datapoint depicts a football match and, as such, contains: the date in which the match took place, which determines the season for which the match was valid; the league; the home team; the away team; the final score (as in, the final amount of goals scored by each team); and the pay-offs at *Bet 365* betting house.

The database encompasses 11 different European countries and their top division leagues (Belgium, England, France, Germany, Greece, Italy, The Netherlands, Portugal, Scotland, Spain and Turkey), ranging from the 1993–1994 to the 2018–2019 seasons (some leagues only have data starting in the 1994–1995 and 1995–1996 seasons). These data were extracted from https://www.football-data.co.uk/. Considering all leagues and seasons, the final database encompasses 87 816 matches (table 3). These 87 816 matches had a total of 236 323 goals scored, an average of 2.7 goals per match.

## 4.2. Modelling predictability

To measure the predictability of football based on a minimum amount of available data, we need to build a simple prediction model. For the sake of simplicity and interpretability of our model, we limit our analysis to the matches that have a winner and we eliminate the ties from the entirety of this study. To include the ties an additional parameter would be needed which makes the comparison between different years (different model tuning) irrelevant. Electronic supplementary material, figure S6 shows

**Table 3.** Database: matches per league.

| Country | Total Matches |
| --- | --- |
| Belgium | 6620 |
| England | 10 044 |
| France | 9510 |
| Germany | 7956 |
| Greece | 6470 |
| Italy | 9066 |
| Netherlands | 7956 |
| Portugal | 7122 |
| Scotland | 5412 |
| Spain | 10 044 |
| Turkey | 7616 |

the percentage of matches ending in a tie is either constant or slightly diminishing throughout time for all but two countries. Our prediction task, therefore, is simplified to predicting a home win versus an away win that refer to the events of the host, or the guest team wins the match respectively. To predict the results of a given match, we consider the performance of each of the two competing teams in their past $N$ matches preceding that given match. To be able to compare leagues with different numbers of teams and matches, we normalize this by the total number of matches played in each season $T$, and define the model training window as $n = N/T$.

For each team, we calculate its accumulated *dyadic score* as the fraction of points the team has earned during the window $n$ to the maximum number of points that the team could win during that window. Using this score as a proxy of the team's strength, we can calculate the difference between the strengths of the two teams prior to their match and train a predictive model which considers this difference as the input.

However, this might be too much of a naive predictive model, given that the two teams have most likely played against different sets of teams and the points that they collected can mean different levels of strength depending on the strength of the teams that they collected the points against. We propose a network-based model in order to improve the naive dyadic model and account for this scenario.

### 4.2.1. Network model

To overcome the above-mentioned limitation and come up with a scoring system that is less sensitive to the set of teams that each team has played against, we build a directed network of all the matches within the training window, in which the edges point from the loser to the winner, weighted by the number of points the winner earned. In the next step, we calculate the network *eigenvector centrality* score for all the teams. The recursive definition of eigenvector centrality, that is that the score of each node depends on the score of its neighbours that send a link to it, perfectly solves the problem of the dyadic scoring system mentioned above [17,32,33]. An example of such network and calculated scores are presented in figure 3.

We can calculate the score difference between the two competing teams for any match after the $N$th match. We will have $(T - N)$ matches with their respective outcomes and *score differences*, for both models. We then fit a *logistic regression model* of the outcome (as a categorical $y$ variable) over the score differences according to equation (4.1). Without loss of generality, we always calculate the point score difference as $x = $ home team score $-$ away team score and assume $y = 1/0$ if the home/away team wins.

$$y = F(x | \mu, s) = \frac{1}{1 + \exp(-(x - \mu/s))},$$  (4.1)

where $\mu$ and $s$ are model parameters than can be obtained by ordinary least-squares methods.

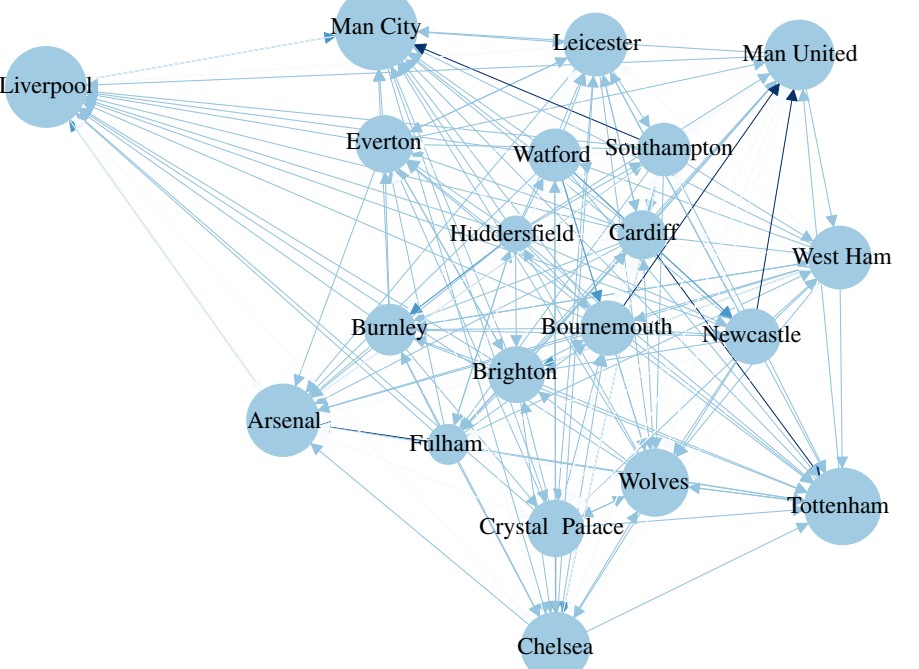

**Figure 3.** The network diagram of the 2018–2019 English Premier League after 240 matches have been played for $n = 0.5$ (calculating centrality scores based on the last 190 matches).

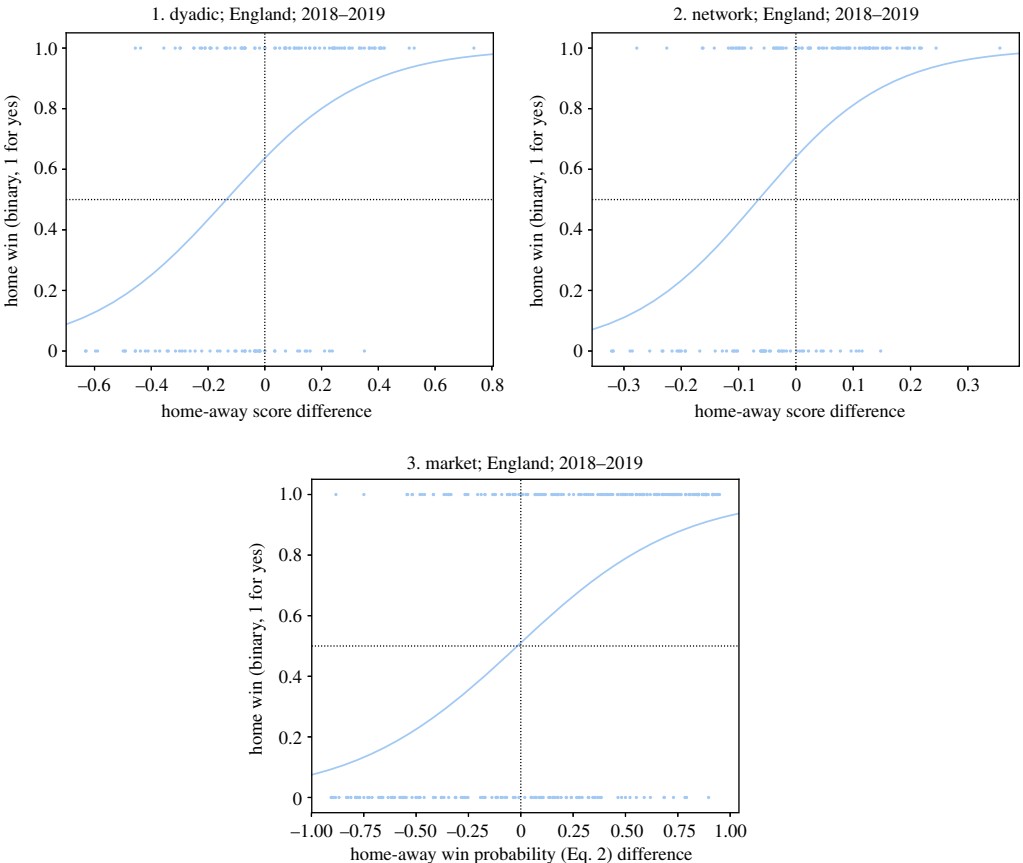

**Figure 4.** Logistic regression model example for the England Premier League 2018–2019 for the two models and the benchmark model.

Finally, the logistic regression model will provide the probabilistic assessment of each score system (dyadic and network) for each match, allowing us to understand how correctly the outcomes are being split as a function of the pre-match score difference. See figure 4 for an example.

### 4.2.2. Quantifying home advantage

In figure 4, we see that the fitted sigmoid is not symmetric and a shift in the probable outcome (i.e. when $\text{Prob}(y = 1) \geq 0.50$) happens at an $x$-value smaller than zero, meaning even for some negative values of $x$, which represents cases where the visiting team's score is greater than the home team's score, we still observe a higher chance for a home team victory. This is not surprising considering the existence of the notion of *home-field advantage*: a significant and structural difference in outcome contingent on the locality in which a sports match is played—favouring the team established in such locality, has been exhaustively uncovered between different team sports and between different countries both for men's and women's competitions [34]. Home-field advantage was first formalized for North-American sports [35] and then documented specifically for association football [36].

### 4.2.3. Benchmark model

We benchmark our models against the implicit prediction model provided by the betting market pay-offs (*Bet 365*). When betting houses allow players to bet on an event outcome, they are essentially providing an implicit probabilistic assessment of that event through the pay-off values. We calculate the probabilities of different outcomes as the following:

$$\left.\begin{array}{r} \text{Prob (home win)} = \dfrac{1/h}{(1/h) + (1/a)} \\[2mm] \text{Prob (away win)} = \dfrac{1/a}{(1/h) + (1/a)} \end{array}\right\}, \tag{4.2}$$

and

where $h$ is monetary units for a home team win, $a$ units for an away team win. The experienced gambler knows that in the real world, on any given betting house's game, the sum of the outcomes' probabilities never does equal to 1, as the game proposed by the betting houses is not strictly *fair*. However, we believe this set of probabilities give us a good estimate of the state-of-the-art predictive models in the market. In figure 4, the data and corresponding fits to all the three models are provided.

### 4.2.4. The Elo ranking system

The problem of ranking teams/players within a tournament based not only on their victories and defeats but also on *how hard those matches were expected to be beforehand* is rather explored throughout the literature. Arpad Elo [37] proposed what became the first ranking system to explicitly consider such caveat, as follows: new players/teams start with a numeric rating ($R_i$) of 1500. Prior to each match we calculate the expected result for each team ($E_i$), which is a quasi-normal function of the difference between the teams' current rankings ($R_i$ and $R_j$)

$$E_i = \frac{1}{1 + 10^{(R_j - R_i)/400}}. \tag{4.3}$$

After the match, we update the rankings given the difference between the expected outcome and the actual one

$$R_i' = R_i + K(S_i - E_i), \tag{4.4}$$

where $S_i$ is the binary outcome of the match between teams/players $i$ and $j$, and $E_i$ is the expected outcome for player/team $i$, as determined by equation (4.3). Note a $K$-factor is proposed, which serves as a weight on how much the result will impact the new rating for the teams/players. Elo himself proposed two values of $K$ when ranking chess players, depending on the player's classification (namely 16 for a chess master and 32 for lower-ranked players). Despite the rating itself being composed of a prediction, we applied the same dynamics under which we analysed the prior models: we calculated the rating for both teams prior to each match and used the difference between ratings (home minus away) to fit a logistic curve (with 1 for a home win, 0 for an away win).

## 4.3. Model performance

Assessing the performance of a probabilistic model is both challenging and controversial in the literature, as the different measures present strengths and weaknesses. Notwithstanding, the majority of research done in the area applies (at least one of) two methods, namely: a loss function; and/or a binary accuracy measure. We analyse our models within both approaches and also benchmark the proposed network model against an established ranking model, known as the *Elo ranking model* (see below).

### 4.3.1. Loss function: the Brier score

Proposed in [38], the Brier score is a loss function that measures the accuracy of probabilistic predictions. It applies to tasks in which predictions must assign probabilities to a set of mutually exclusive discrete outcomes. The set of possible outcomes can be either binary or categorical in nature, and the probabilities assigned to this set of outcomes must sum to one (where each individual probability is in the range of 0 to 1).

Suppose that on each of $t$ occasions an event can occur in only one of $r$ possible classes or categories and on one such occasion, $i$, the forecast probabilities are $f_{i1}, f_{i2}, \ldots, f_{ir}$, that the event will occur in classes 1, 2, ..., $r$, respectively. The $r$ classes are chosen to be mutually exclusive and exhaustive so that $\sum_{j=1}^{r} f_{ij} = 1$, $i = 1, 2, 3, \ldots, t$. Hence, the Brier score is defined as

$$P = \frac{1}{t} \sum_{j=1}^{r} \sum_{i=1}^{t} (f_{ij} - E_{ij})^2, \tag{4.5}$$

where $E_{ij}$ takes the value 1 or 0 according to whether the event occurred in class $j$ or not. In our case, we have: $r = 2$ classes, i.e. either the home team won or not; and $(1 - n)T$ matches to be predicted, transforming equation (4.5) into

$$P = \frac{1}{(1 - n)T} \sum_{j=1}^{2} \sum_{i=1}^{(1-n)T} (f_{ij} - E_{ij})^2. \tag{4.6}$$

We see, from equation (4.6), how the Brier score is an averaged measure over all the predicted matches, i.e. each league–season combination will have its own single Brier score. The lower the Brier score, the better (as in the more assertive and/or correct) our prediction model. In electronic supplementary material, figure S7, the distributions of Brier scores for different models are shown.

### 4.3.2. Accuracy: receiver operating characteristic curve

A *receiver operating characteristic curve*, or ROC curve, is a graphical plot that illustrates the diagnostic ability of a binary classifier system as its discrimination threshold is varied, which allows us to generally assess a prediction method's performance for different thresholds. When using normalized units, the area under the ROC curve (often referred to as simply the AUC) is equal to the probability that a classifier will rank a randomly chosen positive instance higher than a randomly chosen negative one [39].

The machine learning community has historically used this measure for model comparison [40], though the practice has been questioned given AUC estimates are quite noisy and suffer from other problems [41–43]. Nonetheless, the coherence of AUC as a measure of aggregated classification performance has been vindicated, in terms of a uniform rate distribution [44], and AUC has been linked to several other performance metrics, including the above-mentioned Brier score [45]. In electronic supplementary material, figure S8, the distributions of Brier scores for different models are shown.

To compare our two models with the benchmark model, we calculate a loss function in the form of the Brier score for all the $(1 - n)T$ matches in different leagues and years from 2005 to 2018 and $n = 0.5$. The distributions of scores are reported in electronic supplementary material, figure S1 and the average scores are shown in figure 5a (note that the smaller the Brier score, the better fit of the model). Progressing on the loss function to actual prediction accuracy, for each match, we considered a home team win prediction whether Prob(home win) $\geq \pi$ and an away team win prediction otherwise. We applied this prediction rule for all three models (dyadic, network and the betting houses market benchmark) and compared the predictions with the outcomes, calculating how many predictions matched the outcome. Note that the betting market prediction data are only available for more recent years (13 years), therefore our comparison is limited to those years. We change the threshold of $\pi$ and

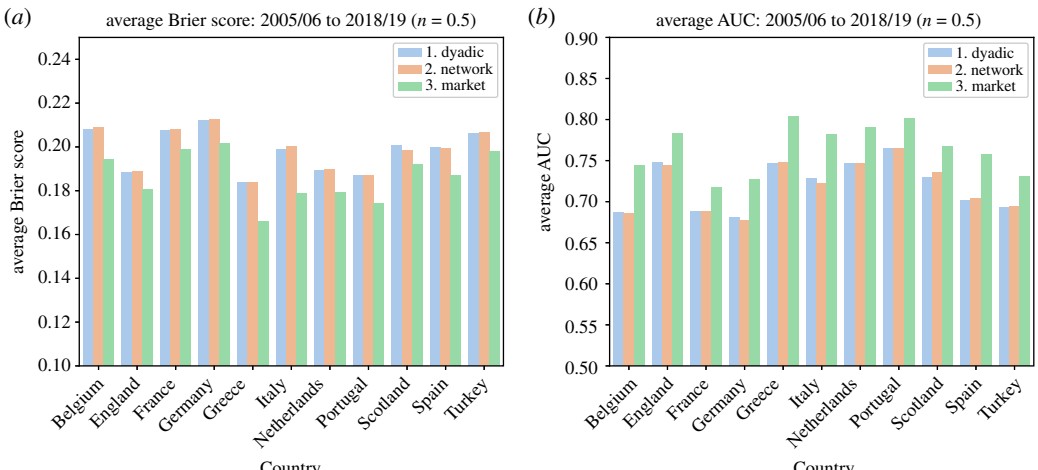

**Figure 5.** Accuracy of the models measured through (*a*) average Brier score and (*b*) area under the curve (AUC), per league.

calculate the average AUC score of ROC curve for each league, shown in figure 5*b*. The complete set of distributions for the accuracy indicators is depicted in electronic supplementary material, figures S1 and S2. By increasing $n$, naturally the accuracy of predictions increases. For the rest of the paper, however, we focus on $n = 0.5$. The comparisons for different values of $n$ are presented in electronic supplementary material, figures S9–S12.

Even though our model underperforms the market on average, $t$-tests for the difference between the Brier score and AUC distributions of our models and the betting houses market are statistically indistinguishable at the 2% significance level for the majority of year-leagues (see electronic supplementary material, tables S1 and S2 for details). One should note that the goal here is not to develop a better prediction model than the one provided by the betting houses market but to obtain a comparable, consistent and simple-to-calculate model. Market models evolve throughout time, which would bias the assessment of a single prediction tool throughout the years. In that capacity, we are looking for models that provide the time-robust tool we need to assess whether football is becoming more predictable over the years.

To compare the network model with the Elo-based model, we calculated the AUC for the Elo-based model for each country–season combination and plotted it against the network model. The results are surprisingly similar, but the network model outperforms an Elo-based model (with a $K$-factor of 32), as we can see from electronic supplementary material, figure S13. Specifically, the network model scores a higher AUC value in roughly 61% of cases (170 of 280). Nevertheless, the results for the trends in predictability hold for the Elo-based model too. However, as said above, what matters the most is the ease in training of the model and the consistency of its predictions rather than slight advantages in performance.

Data accessibility. The datasets used and analysed during the current study are available from the Dryad Digital Repository: https://doi.org/10.5061/dryad.8931zcrrs [46].
Authors' contributions. V.M.M. collected and analysed the data. T.Y. designed and supervised the research. Both authors wrote the article.
Competing interests. We declare we have no competing interests.
Funding. T.Y. was partially supported by the Alan Turing Institute under the EPSRC grant no. EP/N510129/1.
Acknowledgements. The authors thank Luca Pappalardo for valuable discussions.

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
