## [Peer Review File · Royal Society Open Science]

Review History

RSOS-210617.R0 (Original submission)

Review form: Reviewer 1

Is the manuscript scientifically sound in its present form?

No

Are the interpretations and conclusions justified by the results?

No

Is the language acceptable?

Yes

Do you have any ethical concerns with this paper?

No

Have you any concerns about statistical analyses in this paper?

Yes

Recommendation?

Major revision is needed (please make suggestions in comments)

Comments to the Author(s)

This paper proposes the thesis that football is becoming more predictable: a simple model trying to predict a home/away win registers increasing accuracy scores over time. Several hypotheses are considered, for instance a feedback loop in income inequality that makes strong teams stronger over time.

Overall this is a short paper with a precise question and straightforward methodology. This increases its clarity. Moreover, the topic is interesting because it focuses on a popular activity involving strong economic interests and it can provide insights on how to make it more engaging -- and fair -- in the future. Thus it deserves consideration and attention.

I think that the paper needs a few additional touches before being considered of publishable quality.

The most important issue that is front and center is that the authors propose the income inequality hypothesis in the abstract as the driving engine of their result, but income data is never used in the paper. It would be much more convincing if the naive prediction model would use not eigenvector centrality, but team budget ratio. The authors should make an effort to retrieve this data, because it would increase the credibility of their hypotheses.

Leaving aside the need for additional data, eigenvector centrality doesn't seem necessarily the only nor the best choice for a simple predictive model. One could use the ELO score from chess, and/or the world rugby ranking system (<https://www.world.rugby/tournaments/rankings/explanation>), both of which are quite stable and reliable. Having multiple models is also necessary because it's possible that the results might be just a random fluke of a single model: if football is really becoming more predictable then all models should show an accuracy increase, not only one.

Another major issue is Fig 4: it should be the main result of the paper, but I'm hesitant in saying it is anything different than null. By squinting at the figure we could say that some leagues increased their AUCs (predictability), but are the values in the final years really statistically higher than the ones in the early years? (For instance in Scotland or France that are cited as examples of increased predictability) The authors really need to have some objective statistical test to confirm that the >0.75 AUC in France in 1993 is truly lower (?) than the ~ 0.7 AUC of the same league 2018.

Other important, but more defensible, issues are that throwing away ties is suboptimal -- a tie can be a huge upset. Moreover, the home field advantage analysis is an interesting observation, but a bit underdeveloped: there are only hypotheses of what might be the case without any test. Can the authors expand on it?

Minor:

- Some typos ("sore" instead of "score"), and the quality of the text could be improved in general.
- It should be noted in the data section that the betting data is only available for less than half the matches.
- Besides including data from more leagues, as the authors suggest in their discussion, it would be interesting also to look at inter-league competitions such as Champions League and UEFA Cup.

Review form: Reviewer 2 (Rui Marcelino)

Is the manuscript scientifically sound in its present form?

Yes

Are the interpretations and conclusions justified by the results?

No

Is the language acceptable?

Yes

Do you have any ethical concerns with this paper?

No

Have you any concerns about statistical analyses in this paper?

No

Recommendation?

Major revision is needed (please make suggestions in comments)

Comments to the Author(s)

Although it seems to me very relevant and pertinent to conduct a serious and complete investigation as the authors proposed in this work, I believe that it could be improved if some adjustments were made and additional information provided.

So, starting with the beginning, I don't think the proposed title is the most adequate. It doesn't seem to me relevant that the title is composed by a somewhat subjective conclusion from a scientific point of view ("becoming boring"), and by details regarding the methodology... As it is presented, it represents an excessive reduction of the work carried out simultaneously with failing to present essential characteristics of the work. I think I can understand the authors' goals by presenting a more appealing title (commercial, even), but a reformulation is recommended, respecting the imperatives associated with the production and dissemination of scientific research. In the same sense, a total restructuring of the presented abstract is advised, as it leaves out essential aspects of the work developed.

The overall structure of the work would also benefit if the contents presented in the "Modellig predictability" section were all presented within the "Materials and Methods" section. This change will allow for greater fluidity in what will be the reading of the work, leaving more technical issues for further analysis. As presented, there is a rather abrupt transition between the reasoning presented in the introduction (about the state of the art and relevance of conducting this type of research) and the more methodological/mathematical aspects associated with what the authors have done.

According to my understanding, this work presents a huge weakness when it disregards all the games that ended in a draw. Effectively, and being a sport with low scores, ties in football are frequent and represent a very important part of what is the structure of the sport from a competitive point of view. I understand that the method used (probably because it is binary) facilitates the analysis as it was performed (ie, compare home win versus an away win), however it does not represent what is the relevance of the phenomenon under study. I believe that scientific problems should not be adapted to the available statistical methods, but the opposite: in other words, faced with a problem (in this case, the importance of understanding/measure the predictability of football, in which there are three possible outcomes: victory, draw and defeat),

statistical methods must be found that effectively contribute to clarifying the existing relationships in the phenomena under study.

Thus, it is necessary to redo the analyzes so that the ties are considered in this study.

In several sessions of the manuscript, associations with machine learning techniques emerge. However, it is not clear whether the authors used, or intended to use, this type of technique. Not having used them (as it seems to me) I think it would be convenient to provide additional explanations associated with these techniques...if they are considered relevant and useful to the point of being frequently mentioned, why have they not been applied with the available database? (which, by the way, is very well prepared); in case you have applied some technique (which I don't think has happened), there is a lot of information to clarify: algorithm used, training and testing samples....

In summary, the work has the potential to contribute positively to the knowledge associated with performance in this modality over the last decade, but it needs to be strongly restructured. It is recommended to include all games that ended with draws and reorganize the text so that the message is conveyed more effectively...

Rui Marcelino

16/08/2021

Decision letter (RSOS-210617.R0)

Dear Professor Yasseri

The Editors assigned to your paper RSOS-210617 "Football is becoming boring; Network analysis of 88 thousands matches in 11 major leagues" have now received comments from reviewers and would like you to revise the paper in accordance with the reviewer comments and any comments from the Editors. Please note this decision does not guarantee eventual acceptance.

Please submit your revised manuscript and required files (see below) no later than 21 days from today's (ie 02-Sep-2021) date. Note: the ScholarOne system will 'lock' if submission of the revision is attempted 21 or more days after the deadline. If you do not think you will be able to meet this deadline please contact the editorial office immediately.

Please note article processing charges apply to papers accepted for publication in Royal Society Open Science (<https://royalsocietypublishing.org/rsos/charges>). Charges will also apply to

papers transferred to the journal from other Royal Society Publishing journals, as well as papers submitted as part of our collaboration with the Royal Society of Chemistry (<https://royalsocietypublishing.org/rsos/chemistry>). Fee waivers are available but must be requested when you submit your revision (<https://royalsocietypublishing.org/rsos/waivers>).

on behalf of Dr Mirco Musolesi (Associate Editor) and Marta Kwiatkowska (Subject Editor)
openscience@royalsociety.org

Associate Editor Comments to Author (Dr Mirco Musolesi):

Associate Editor: 1

Comments to the Author:

The reviewers raised some concerns that should be addressed by the authors in a revised version of the paper. I would suggest to carefully address all the points raised by the authors, but in particular, the following issues require particular attention:

- "income inequality" is mentioned, but not actually used. Please add an analysis using these data if possible.
- compare with ELO-like scores or other standard scores;
- explain Fig 4.: it seems that the results are actually null;
- add drawings to your model, since they are a very relevant case.
- there are many references to machine learning techniques, but it appears that they have not actually been used? The authors should provide more details about this.

Associate Editor: 2

Comments to the Author:

The paper will be sent for peer review.

Reviewer comments to Author:

Reviewer: 1

Comments to the Author(s)

This paper proposes the thesis that football is becoming more predictable: a simple model trying to predict a home/away win registers increasing accuracy scores over time. Several hypotheses are considered, for instance a feedback loop in income inequality that makes strong teams stronger over time.

Overall this is a short paper with a precise question and straightforward methodology. This increases its clarity. Moreover, the topic is interesting because it focuses on a popular activity

involving strong economic interests and it can provide insights on how to make it more engaging -- and fair -- in the future. Thus it deserves consideration and attention.

I think that the paper needs a few additional touches before being considered of publishable quality.

The most important issue that is front and center is that the authors propose the income inequality hypothesis in the abstract as the driving engine of their result, but income data is never used in the paper. It would be much more convincing if the naive prediction model would use not eigenvector centrality, but team budget ratio. The authors should make an effort to retrieve this data, because it would increase the credibility of their hypotheses.

Leaving aside the need for additional data, eigenvector centrality doesn't seem necessarily the only nor the best choice for a simple predictive model. One could use the ELO score from chess, and/or the world rugby ranking system (<https://www.world.rugby/tournaments/rankings/explanation>), both of which are quite stable and reliable. Having multiple models is also necessary because it's possible that the results might be just a random fluke of a single model: if football is really becoming more predictable then all models should show an accuracy increase, not only one.

Another major issue is Fig 4: it should be the main result of the paper, but I'm hesitant in saying it is anything different than null. By squinting at the figure we could say that some leagues increased their AUCs (predictability), but are the values in the final years really statistically higher than the ones in the early years? (For instance in Scotland or France that are cited as examples of increased predictability) The authors really need to have some objective statistical test to confirm that the >0.75 AUC in France in 1993 is truly lower (?) than the ~ 0.7 AUC of the same league 2018.

Other important, but more defensible, issues are that throwing away ties is suboptimal -- a tie can be a huge upset. Moreover, the home field advantage analysis is an interesting observation, but a bit underdeveloped: there are only hypotheses of what might be the case without any test. Can the authors expand on it?

Minor:

- Some typos ("sore" instead of "score"), and the quality of the text could be improved in general.
- It should be noted in the data section that the betting data is only available for less than half the matches.
- Besides including data from more leagues, as the authors suggest in their discussion, it would be interesting also to look at inter-league competitions such as Champions League and UEFA Cup.

Reviewer: 2

Comments to the Author(s)

Although it seems to me very relevant and pertinent to conduct a serious and complete investigation as the authors proposed in this work, I believe that it could be improved if some adjustments were made and additional information provided.

So, starting with the beginning, I don't think the proposed title is the most adequate. It doesn't seem to me relevant that the title is composed by a somewhat subjective conclusion from a scientific point of view ("becoming boring"), and by details regarding the methodology... As it is presented, it represents an excessive reduction of the work carried out simultaneously with failing to present essential characteristics of the work. I think I can understand the authors' goals

by presenting a more appealing title (commercial, even), but a reformulation is recommended, respecting the imperatives associated with the production and dissemination of scientific research. In the same sense, a total restructuring of the presented abstract is advised, as it leaves out essential aspects of the work developed.

The overall structure of the work would also benefit if the contents presented in the "Modelling predictability" section were all presented within the "Materials and Methods" section. This change will allow for greater fluidity in what will be the reading of the work, leaving more technical issues for further analysis. As presented, there is a rather abrupt transition between the reasoning presented in the introduction (about the state of the art and relevance of conducting this type of research) and the more methodological/mathematical aspects associated with what the authors have done.

According to my understanding, this work presents a huge weakness when it disregards all the games that ended in a draw. Effectively, and being a sport with low scores, ties in football are frequent and represent a very important part of what is the structure of the sport from a competitive point of view. I understand that the method used (probably because it is binary) facilitates the analysis as it was performed (ie, compare home win versus an away win), however it does not represent what is the relevance of the phenomenon under study. I believe that scientific problems should not be adapted to the available statistical methods, but the opposite: in other words, faced with a problem (in this case, the importance of understanding/measure the predictability of football, in which there are three possible outcomes: victory, draw and defeat), statistical methods must be found that effectively contribute to clarifying the existing relationships in the phenomena under study.

Thus, it is necessary to redo the analyzes so that the ties are considered in this study.

In several sessions of the manuscript, associations with machine learning techniques emerge. However, it is not clear whether the authors used, or intended to use, this type of technique. Not having used them (as it seems to me) I think it would be convenient to provide additional explanations associated with these techniques...if they are considered relevant and useful to the point of being frequently mentioned, why have they not been applied with the available database? (which, by the way, is very well prepared); in case you have applied some technique (which I don't think has happened), there is a lot of information to clarify: algorithm used, training and testing samples...

In summary, the work has the potential to contribute positively to the knowledge associated with performance in this modality over the last decade, but it needs to be strongly restructured. It is recommended to include all games that ended with draws and reorganize the text so that the message is conveyed more effectively...

Rui Marcelino
16/08/2021

===PREPARING YOUR MANUSCRIPT===

Your revised paper should include the changes requested by the referees and Editors of your manuscript. You should provide two versions of this manuscript and both versions must be provided in an editable format:
one version identifying all the changes that have been made (for instance, in coloured highlight, in bold text, or tracked changes);
a 'clean' version of the new manuscript that incorporates the changes made, but does not highlight them. This version will be used for typesetting if your manuscript is accepted.

===PREPARING YOUR REVISION IN SCHOLARONE===

- If you are providing image files for potential cover images, please upload these at this step, and inform the editorial office you have done so. You must hold the copyright to any image provided.
- A copy of your point-by-point response to referees and Editors. This will expedite the preparation of your proof.

- Ensure that your data access statement meets the requirements at <https://royalsociety.org/journals/authors/author-guidelines/#data>. You should ensure that you cite the dataset in your reference list. If you have deposited data etc in the Dryad repository, please include both the 'For publication' link and 'For review' link at this stage.
- If you are requesting an article processing charge waiver, you must select the relevant waiver option (if requesting a discretionary waiver, the form should have been uploaded at Step 3 'File upload' above).
- If you have uploaded ESM files, please ensure you follow the guidance at <https://royalsociety.org/journals/authors/author-guidelines/#supplementary-material> to include a suitable title and informative caption. An example of appropriate titling and captioning may be found at [https://figshare.com/articles/Table_S2_from_Is_there_a_trade-off_between_peak_performance_and_performance_breadth_across_temperatures_for_aerobic_sc](https://figshare.com/articles/Table_S2_from_Is_there_a_trade-off_between_peak_performance_and_performance_breadth_across_temperatures_for_aerobic_scope_in_teleost_fishes_/3843624) ope_in_teleost_fishes_/3843624.

Author's Response to Decision Letter for (RSOS-210617.R0)

See Appendix A.

RSOS-210617.R1

Review form: Reviewer 1

Is the manuscript scientifically sound in its present form?

Yes

Are the interpretations and conclusions justified by the results?

Yes

Is the language acceptable?

Yes

Do you have any ethical concerns with this paper?

No

Have you any concerns about statistical analyses in this paper?

No

Recommendation?

Accept with minor revision (please list in comments)

Comments to the Author(s)

The authors addressed all my comments in a satisfactory manner and I think the paper should be published now.

A few remarks that I think they still should consider:

- Use some sort of visual aid (perhaps boldface) in Table 1 to distinguish the significant from the non-significant p-value for a standard threshold (0.05?). This would make it easier to parse the results and distinguish the leagues that became more predictable from the ones that did not.

- It doesn't really matter which one between ELO or the network model (or the betting market, for that matter) achieves a higher AUC: the important thing is that the AUCs behave in similar ways for all predicting models in the same leagues -- which they do, as far as I can tell. So discussing which one "performs best" could distract the reader from what's important.

Review form: Reviewer 2 (Rui Marcelino)**Is the manuscript scientifically sound in its present form?**

Yes

Are the interpretations and conclusions justified by the results?

Yes

Is the language acceptable?

Yes

Do you have any ethical concerns with this paper?

No

Have you any concerns about statistical analyses in this paper?

No

Recommendation?

Accept with minor revision (please list in comments)

Comments to the Author(s)

I congratulate the authors for the changes made, which, in my opinion, favor the reading and interpretation of the work developed.

As a small issue, I recommend that the figures will be revised with regard to the y-axis scale:

When there are figures with sub-plots, which intend to represent the different Leagues, it is convenient that the scale to be used is identical in all sub-plots. The standardization of the scale by all Leagues will guarantee a more correct interpretation of all presented results.

There is need to do this in Figure 1 (left and right axis); Fig. S1;

Fig. S2; Fig. S3; Fig. S4; Fig. S5; Fig. S6; Fig. S7; Fig. S8; Fig. S13; Fig. S14

As a small detail, it is recommended to standardize the number of decimal places to be used in the figures. In some cases one is used, in others 2 and 3 other decimal places.

Decision letter (RSOS-210617.R1)

Dear Professor Yasseri,

On behalf of the Editors, we are pleased to inform you that your Manuscript RSOS-210617.R1 "Football is becoming more predictable; Network analysis of 88 thousands matches in 11 major leagues" has been accepted for publication in Royal Society Open Science subject to minor revision in accordance with the referees' reports. Please find the referees' comments along with any feedback from the Editors below my signature.

Please submit your revised manuscript and required files (see below) no later than 7 days from today's (ie 19-Nov-2021) date. Note: the ScholarOne system will 'lock' if submission of the revision is attempted 7 or more days after the deadline. If you do not think you will be able to meet this deadline please contact the editorial office immediately.

on behalf of Dr Mirco Musolesi (Associate Editor) and Marta Kwiatkowska (Subject Editor)
openscience@royalsociety.org

Associate Editor Comments to Author (Dr Mirco Musolesi):

The authors addressed all the concerns of the reviewers in a convincing manner. I would like to recommend this manuscript for publication.

There are very minor stylistic issues highlighted by the reviewers which the authors might want to address in the preparation of the camera ready version of the paper.

Reviewer comments to Author:

Reviewer: 1

Comments to the Author(s)

The authors addressed all my comments in a satisfactory manner and I think the paper should be published now.

A few remarks that I think they still should consider:

- Use some sort of visual aid (perhaps boldface) in Table 1 to distinguish the significant from the non-significant p-value for a standard threshold (0.05?). This would make it easier to parse the results and distinguish the leagues that became more predictable from the ones that did not.

- It doesn't really matter which one between ELO or the network model (or the betting market, for that matter) achieves a higher AUC: the important thing is that the AUCs behave in similar ways for all predicting models in the same leagues -- which they do, as far as I can tell. So discussing which one "performs best" could distract the reader from what's important.

Reviewer: 2

Comments to the Author(s)

I congratulate the authors for the changes made, which, in my opinion, favor the reading and interpretation of the work developed.

As a small issue, I recommend that the figures will be revised with regard to the y-axis scale: When there are figures with sub-plots, which intend to represent the different Leagues, it is convenient that the scale to be used is identical in all sub-plots. The standardization of the scale by all Leagues will guarantee a more correct interpretation of all presented results.

There is need to do this in Figure 1 (left and right axis); Fig. S1;

Fig. S2; Fig. S3; Fig. S4; Fig. S5; Fig. S6; Fig. S7; Fig. S8; Fig. S13; Fig. S14

As a small detail, it is recommended to standardize the number of decimal places to be used in the figures. In some cases one is used, in others 2 and 3 other decimal places.

===PREPARING YOUR MANUSCRIPT===

one version should clearly identify all the changes that have been made (for instance, in coloured highlight, in bold text, or tracked changes);

Please ensure that you include an acknowledgements' section before your reference list/bibliography. This should acknowledge anyone who assisted with your work, but does not

qualify as an author per the guidelines at <https://royalsociety.org/journals/ethics-policies/openness/>.

===PREPARING YOUR REVISION IN SCHOLARONE===

- Ensure that your data access statement meets the requirements at <https://royalsociety.org/journals/authors/author-guidelines/#data>. You should ensure that you cite the dataset in your reference list. If you have deposited data etc in the Dryad repository, please only include the 'For publication' link at this stage. You should remove the 'For review' link.
- If you are requesting an article processing charge waiver, you must select the relevant waiver option (if requesting a discretionary waiver, the form should have been uploaded, see 'File upload' above).
- If you have uploaded any electronic supplementary (ESM) files, please ensure you follow the guidance at <https://royalsociety.org/journals/authors/author-guidelines/#supplementary-material> to include a suitable title and informative caption. An example of appropriate titling and captioning may be found at https://figshare.com/articles/Table_S2_from_Is_there_a_trade-off_between_peak_performance_and_performance_breadth_across_temperatures_for_aerobic_scope_in_teleost_fishes_/3843624.

Author's Response to Decision Letter for (RSOS-210617.R1)

See Appendix B.

Decision letter (RSOS-210617.R2)

Dear Professor Yasseri,

I am pleased to inform you that your manuscript entitled "Football is becoming more predictable; Network analysis of 88 thousands matches in 11 major leagues" is now accepted for publication in Royal Society Open Science.

on behalf of Dr Mirco Musolesi (Associate Editor) and Marta Kwiatkowska (Subject Editor)
openscience@royalsociety.org

Appendix A

We thank the Associate Editors and the Reviewers for their valuable comments, which we considered and addressed very carefully in revising the manuscript. We believe the revised manuscript is much stronger and therefore are grateful for all the recommendations and suggestions.

Please see our detailed responses to each comment below. We highlighted the major changes in the manuscript by red font colour.

*Best wishes
VMM & TY.*

Associate Editor Comments to Author (Dr Mirco Musolesi):

Associate Editor: 1

Comments to the Author:

The reviewers raised some concerns that should be addressed by the authors in a revised version of the paper. I would suggest to carefully address all the points raised by the authors, but in particular, the following issues require particular attention:

We thank the Associate Editor for the summary. Here we give short summary of the changes we made and below come more detailed responses to each reviewer.

- "income inequality" is mentioned, but not actually used. Please add an analysis using these data if possible.

We added some fresh data and analysis on the trends in income in different leagues. However, the data (particularly historical data) on individual teams is sparse and we did not manage to find any cohesive dataset. To compensate for this and to balance with the data we did manage to analyse and add to the paper, we shorten the reference inequality.

- compare with ELO-like scores or other standard scores;

We did include calculation and comparing with ELO system in the revision. We reported that our model outperforms ELO-based prediction even though it's simpler in implementation and does not require any free parameter optimization. In any account, the historical trends we report, hold for an ELO-based model too (added to the paper).

- explain Fig 4.: it seems that the results are actually null;

We added some statistical tests to show that the results are not null. The increase in predictability is absent/not significant in smaller leagues, but present and statistically significant in big leagues in a match with our hypothesis.

- add drawings to your model, since they are a very relevant case.

This is a valid suggestion, however, including the ties (draws) in the model will add a free parameter in the model and that makes historical comparison irrelevant as for each year we will be fitting a different model. However, to address the comment, we did some analysis on the draws and added them to the paper as well as more description of our motivation to exclude the ties.

- there are many references to machine learning techniques, but it appears that they have not actually been used? The authors should provide more details about this.

This is a very good point! The strength of our work is in the fact that it's not data-thirsty and allows us to perform retrospective analysis in times that the systematic data collection had not been possible! We do refer to data-thirsty Machine Learning models to make this point; to create a contrast. However, this hasn't been very clear in the original manuscript. We tried to clarify this better in the revision.

Associate Editor: 2

Comments to the Author:

The paper will be sent for peer review.

Reviewer comments to Author:

Reviewer: 1

Comments to the Author(s)

This paper proposes the thesis that football is becoming more predictable: a simple model trying to predict a home/away win registers increasing accuracy scores over time. Several hypotheses are considered, for instance a feedback loop in income inequality that makes strong teams stronger over time.

Overall this is a short paper with a precise question and straightforward methodology. This increases its clarity. Moreover, the topic is interesting because it focuses on a popular activity involving strong economic interests and it can provide insights on how to make it more engaging -- and fair -- in the future. Thus it deserves consideration and attention.

We thank the Reviewer for the positive words and accurate summary of our work.

I think that the paper needs a few additional touches before being considered of publishable quality.

The most important issue that is front and center is that the authors propose the income inequality hypothesis in the abstract as the driving engine of their result, but income data is never used in the paper. It would be much more convincing if the naive prediction model would

use not eigenvector centrality, but team budget ratio. The authors should make an effort to retrieve this data, because it would increase the credibility of their hypotheses.

This is a very relevant point. We did try to build a model entirely based on expenditure/income or other financial indicators, however, such data are simply not available at the club level and for sure not for a long enough time period. However, to address the comment to the best we could, we acquired some league level data and did some preliminary analysis on them and added them to the paper. We hope this gives enough support for our argument, or better put, enough motivation for future research.

Leaving aside the need for additional data, eigenvector centrality doesn't seem necessarily the only nor the best choice for a simple predictive model. One could use the ELO score from chess, and/or the world rugby ranking system (<https://www.world.rugby/tournaments/rankings/explanation>), both of which are quite stable and reliable. Having multiple models is also necessary because it's possible that the results might be just a random fluke of a single model: if football is really becoming more predictable then all models should show an accuracy increase, not only one.

We thank the Reviewer for drawing our attention to this. We implemented the ELO system and redid all the calculations. The results are added to the paper, but in short, our model outperforms ELO even though it has fewer free parameters. Nevertheless, we believe this has made the paper stronger as similar historical trends are revealed by the ELO method too. Thanks for the suggestion.

Another major issue is Fig 4: it should be the main result of the paper, but I'm hesitant in saying it is anything different than null. By squinting at the figure we could say that some leagues increased their AUCs (predictability), but are the values in the final years really statistically higher than the ones in the early years? (For instance in Scotland or France that are cited as examples of increased predictability) The authors really need to have some objective statistical test to confirm that the >0.75 AUC in France in 1993 is truly lower (?) than the ~ 0.7 AUC of the same league 2018.

Following this, we did some statistical tests to check if the increased claim in certain leagues is statistically significant. The results of the tests are in the revised version. In short in smaller leagues, the increase is not significant, but in larger leagues (more income) the increase is significant. Thank you for the suggestion again.

Other important, but more defensible, issues are that throwing away ties is suboptimal -- a tie can be a huge upset.

We intentionally didn't bring the ties in the analysis as they add a free parameter to the model meaning that each year we will be training a different model and that makes the historical

comparisons irrelevant. However, to address this comment, we did some analysis on the tie statistics and how they have decreased or remained stable over time. Now we discuss this in more detail in the paper.

Moreover, the home field advantage analysis is an interesting observation, but a bit underdeveloped: there are only hypotheses of what might be the case without any test. Can the authors expand on it?

This is a great line of enquiry. It is hard to test some of our suggestions based on the available data, however, we hope that our work and the "natural" experiment of the Pandemic which made leagues go fan-free for almost a season, inspire future research by us and others.

Minor:

- Some typos ("sore" instead of "score"), and the quality of the text could be improved in general.

Thank you for pointing out this. We gave a paper a careful read and fixed the typos as much as we could.

- It should be noted in the data section that the betting data is only available for less than half the matches.

Done.

- Besides including data from more leagues, as the authors suggest in their discussion, it would be interesting also to look at inter-league competitions such as Champions League and UEFA Cup.

Absolutely, and this is something we are planning to do in a bigger project hopefully leading to a forthcoming paper. We now mention this in the current paper as a suggestion for future work.

We thank you again for all the comments which helped us improve the paper.

Reviewer: 2

Comments to the Author(s)

Although it seems to me very relevant and pertinent to conduct a serious and complete investigation as the authors proposed in this work, I believe that it could be improved if some adjustments were made and additional information provided.

So, starting with the beginning, I don't think the proposed title is the most adequate. It doesn't seem to me relevant that the title is composed by a somewhat subjective conclusion from a scientific point of view ("becoming boring"), and by details regarding the methodology... As it is presented, it represents an excessive reduction of the work carried out simultaneously with

failing to present essential characteristics of the work. I think I can understand the authors' goals by presenting a more appealing title (commercial, even), but a reformulation is recommended, respecting the imperatives associated with the production and dissemination of scientific research. In the same sense, a total restructuring of the presented abstract is advised, as it leaves out essential aspects of the work developed.

We thank the Reviewer for the valid comments. Following this advice, we revised the title and the abstract to introduce the work more accurately.

The overall structure of the work would also benefit if the contents presented in the "Modelling predictability" section were all presented within the "Materials and Methods" section. This change will allow for greater fluidity in what will be the reading of the work, leaving more technical issues for further analysis. As presented, there is a rather abrupt transition between the reasoning presented in the introduction (about the state of the art and relevance of conducting this type of research) and the more methodological/mathematical aspects associated with what the authors have done.

This is a very helpful recommendation. We did follow this and restructured the sections. Now all the technical parts are in the Methods section.

According to my understanding, this work presents a huge weakness when it disregards all the games that ended in a draw. Effectively, and being a sport with low scores, ties in football are frequent and represent a very important part of what is the structure of the sport from a competitive point of view. I understand that the method used (probably because it is binary) facilitates the analysis as it was performed (ie, compare home win versus an away win), however it does not represent what is the relevance of the phenomenon under study. I believe that scientific problems should not be adapted to the available statistical methods, but the opposite: in other words, faced with a problem (in this case, the importance of understanding/measure the predictability of football, in which there are three possible outcomes: victory, draw and defeat), statistical methods must be found that effectively contribute to clarifying the existing relationships in the phenomena under study. Thus, it is necessary to redo the analyzes so that the ties are considered in this study.

This is a very important point. In our analysis, however, considering draws will add a free parameter to the model which we will need to tune separately for each season. Practically that makes it impossible to compare the models of different years and discuss the level of their predictability. The quality of tuning that parameter could introduce biases in our historical analysis. However, to address the comment to the best of our ability without altering the overall design, we did analyse the draws and their overall trends and added the analysis to the revised paper and discuss our motivations for the inclusion of ties in more detail.

In several sessions of the manuscript , associations with machine learning techniques emerge. However, it is not clear whether the authors used, or intended to use, this type of technique. Not having used them (as it seems to me) I think it would be convenient to provide additional explanations associated with these techniques...if they are considered relevant and useful to the point of being frequently mentioned, why have they not been applied with the available database? (which, by the way, is very well prepared); in case you have applied some technique (which I don't think has happened), there is a lot of information to clarify: algorithm used, training and testing samples....

We mention the data-thirsty Machine Learning approaches to contrast them to our model that does not rely on too many data points about each match. The strength of our model therefore is that it can be used historically and to analyse data from the time where several thousands of data points for every match were not generated. We took the reviewer's comment and revised the manuscript to make this point clearer.

In summary, the work has the potential to contribute positively to the knowledge associated with performance in this modality over the last decade, but it needs to be strongly restructured. It is recommended to include all games that ended with draws and reorganize the text so that the message is conveyed more effectively...

Rui Marcelino
16/08/2021

We thank the reviewer for the constructive comments which we followed to the best of our abilities. We hope the changes made has brought the paper to a satisfying level of quality.

Appendix B

Associate Editor Comments to Author (Dr Mirco Musolesi):

The authors addressed all the concerns of the reviewers in a convincing manner. I would like to recommend this manuscript for publication. There are very minor stylistic issues highlighted by the reviewers which the authors might want to address in the preparation of the camera ready version of the paper.

*** We thank the Associate Editor. We addressed all the issues below.***

Reviewer comments to Author:

Reviewer: 1

Comments to the Author(s)

The authors addressed all my comments in a satisfactory manner and I think the paper should be published now. A few remarks that I think they still should consider:

- Use some sort of visual aid (perhaps boldface) in Table 1 to distinguish the significant from the non-significant p-value for a standard threshold (0.05?). This would make it easier to parse the results and distinguish the leagues that became more predictable from the ones that did not.

*** Thank you for your kind words and the comment. We have done this now.***

- It doesn't really matter which one between ELO or the network model (or the betting market, for that matter) achieves a higher AUC: the important thing is that the AUCs behave in similar ways for all predicting models in the same leagues -- which they do, as far as I can tell. So discussing which one "performs best" could distract the reader from what's important.

*** We added a sentence to highlight this.***

Reviewer: 2

Comments to the Author(s)

I congratulate the authors for the changes made, which, in my opinion, favor the reading and interpretation of the work developed.

As a small issue, I recommend that the figures will be revised with regard to the y-axis scale: When there are figures with sub-plots, which intend to represent the different Leagues, it is convenient that the scale to be used is identical in all sub-plots. The standardization of the scale by all Leagues will guarantee a more correct interpretation of all presented results.

There is need to do this in Figure 1 (left and right axis); Fig. S1;

Fig. S2; Fig. S3; Fig. S4; Fig. S5; Fig. S6; Fig. S7; Fig. S8; Fig. S13; Fig. S14.

*** Thank you very much. We have done this now.***

As a small detail, it is recommended to standardize the number of decimal places to be used in the figures. In some cases one is used, in others 2 and 3 other decimal places.

*** Likewise, done.***